# Reducing the Weakening Effect in Fibre-Reinforced Polymers Caused by Integrated Film Sensors

**Alexander Kyriazis** [1,*,†] **, Julia Feder** [1] **, Korbinian Rager** [2,†] **, Chresten von der Heide** [2] **, Andreas Dietzel** [2] **and Michael Sinapius** [1]

1   Institut für Mechanik und Adaptronik, Technische Universität Braunschweig, 38106 Braunschweig, Germany; j.feder@tu-braunschweig.de (J.F.); m.sinapius@tu-braunschweig.de (M.S.)

2   Institut für Mikrotechnik, Technische Universität Braunschweig, 38124 Braunschweig, Germany; k.rager@tu-braunschweig.de (K.R.); c.von-der-heide@tu-braunschweig.de (C.v.d.H.); a.dietzel@tu-braunschweig.de (A.D.)

*   Correspondence: a.kyriazis@tu-braunschweig.de

†   These authors contributed equally to this work.

**Abstract:** Integrating foil sensors into fibre-reinforced plastics offers the advantage of making manufacturing measurable with spatial resolution and thus simplifies quality control. One challenge here is the possible negative influence of the integrated sensors on the mechanical behaviour of the structure. This article shows how the different parts of a film sensor influence important mechanical strength parameters of fibre composites. A comparison of two thermoplastic carrier films shows that by choosing polyetherimide (PEI) instead of polyimide (PI), a considerably more advantageous failure behaviour of the composite is achieved. While integrated PI films reduce the interlaminar shear strength by 68%, no impairment is noticeable due to PEI films. For the critical energy release rate, PEI-based film sensors even lead to a significant increase, while a significant deterioration of 85% can be observed for PI-based sensors. However, not only the film substrate plays a decisive role for the interlaminar shear strength, but also the sensor structures themselves. In this article, sensor structures made of gold were investigated. The decisive parameter for the impairment seems to be the area share of gold structures in the sensor. For a sensor pattern made of gold lines with an area filling of 50%, a reduction of the interlaminar shear strength of up to 25% was observed depending on the angle between the shear stress and the gold lines. No impairment was observed for sensor structures with less gold area. The results show that PEI substrates can be a superior alternative for sensor integration into fibre composites and suggest that there is a trade-off between sensitivity and degradation of mechanical properties when designing interdigital sensors.

**Keywords:** sensor integration; fibre reinforced polymer; FRP; composite; multifunctional composite; interlaminar shear strength; ILS; adhesion; critical energy release rate; design of foil based sensors; foil sensor; interdigital sensor





## 1. Introduction

There are many reasons for using integrated sensors to investigate the curing process of fibre-reinforced plastics. First, curing monitoring is useful for quality assurance in the production of safety-critical parts, in particular, by using sensors to ensure that the components have reached their final strength. However, accelerating production can also be a motivation for curing monitoring, because if complete curing can be reliably detected, excessively long curing cycles can be shortened and cycle times saved. In the infusion process, integrated sensors can not only be used to determine the degree of cure, because a sufficiently dense network additionally allows the detection of the flow front [1] in advance of the curing reaction. A comparatively new topic is represented by so-called smart cure cycles [2,3], in which residual stress states are to be reduced by lowering the temperature of the curing component when the gel point is reached. In this way, a stress-free temperature

closer to the actual service temperature of the component is achieved and residual stresses decrease while the load-bearing capacity of the component is increased. While Kim et al. still predicted the gel point using simulations based on DSC data [3], Prussak et al. instead used integrated fibre Bragg sensors [2], which determine the point at which a mechanical interaction between the epoxy resin and the surrounding laminate begins.

As many properties of the matrix resin simultaneously change during the curing reaction, there is a variety of conceivable observation methods. These can be divided into three categories: procedures in a laboratory environment and procedures with tool-integrated or component-integrated sensors. The classical thermal and thermo-mechanical methods such as dynamic mechanical analysis, differential scanning calorimetry and rheological testing are only suitable for curing measurements in the laboratory environment, but often allow to calculate a simulation model for the curing behaviour of the resin under consideration. A classical tool-integrated method is the measurement of the acoustic properties by means of ultrasonic sensors, especially piezoelectric ceramics [4]. Pommer proved that ultrasonic methods can even be used in the harsh environment of a pultrusion machine for measuring the curing reaction [5]. Fiber Bragg gratings are used as component-integrated sensors for strain measurement [2], but strain gages have also been used to measure the strains occurring during the curing reaction [6].

Furthermore, dielectric sensors are often used as component integrated or component applied sensors [1,7–11]. This type of sensor measures the complex permittivity, which is very sensitive to the progression of the curing reaction. A method for measuring the dielectric properties in curing epoxy resins was already described by Delmonte in 1959 [12]. Maistros et al. related the degree of cure to the frequency response of the electrical impedance already in 1994 [13]. In the meantime, dielectric sensors have made it into commercial applications, such as the curing measurement systems from the Netzsch company [14]. While in 1959 two thin aluminium foils were used for dielectric measurement, forming a plate capacitor, nowadays mainly interdigitated coplanar structures are used [1,7–11]. However, these sensors are often only applied to the surface of a laminate and not integrated between the fibre layers [8].

The integration of sensors between the individual layers of a fibre composite involves many opportunities as well as risks. One opportunity, for example, is that the integration of sensors between different layers allows a thickness-resolved curing measurement in thick laminates, which is not possible with sensors on the surface. One risk is a weakening of the laminate, referred to by Dumstorff et al. as the 'wound effect' [15]. For example, inserting sensors on very stiff and brittle substrates such as glass causes the sensor substrate to break due to its low ductility [16], resulting in a crack inside the laminate that can propagate further through the component. This can be overcome by using sensors on flexible substrates, which, in addition to being less prone to cracking, also reduce the stiffness jump and resulting excess stress. Polyimide (PI) is often used as a flexible substrate [10,16–19], which has high strength and a high thermal resistance. However, this material has the disadvantage of exhibiting a weak bond to the laminate and thus favouring delamination [16].

In order to preserve the advantage of a flexible substrate while reducing the disadvantage of weak bonding, sensors based on plasticised RTM6 were used [11], achieving a significant improvement in adhesion but not yet reaching the level of the laminate without sensors. Another approach considers miniaturising the sensors and providing holes in the sensor to allow cross-linking of the epoxy resin through the sensors [9,16]. These approaches also lead to an improvement in the mechanical properties of the laminate, but do not achieve the strength of the laminate without integrated sensors. Furthermore, the miniaturisation of the sensors does not change the fact that the bond is locally weakened, so that cracks could develop in the vicinity of miniaturised sensors under dynamic stress, which subsequently grow.

To reduce the 'wound effect', the authors use an approach that was proposed by Bruckbauer et al. in 2018 for the bonding of functional layers [20] but has not been tested for integrated sensors before. Instead of PI, thermoplastics are used that can be dissolved

in epoxy resin and thus form an interphase [21–23]. In preliminary investigations, the thermoplastics polyetherimide (PEI) and polyethersulfone, among others, were examined for this purpose [24,25] and PEI was selected on the basis of the preliminary investigations. PEI has the favourable properties of having a modulus of elasticity similar to that of most epoxy resins, of being able to be spin-coated after dissolution in trichloroethanol, of having sufficient elongation at break and thermal resistance, and of having a strong bond to epoxy resin even without pronounced interphase formation [24]. PEI thus has the decisive advantage over PI that it does not promote delamination due to its strong adhesion.

Foil-based sensors, however, also contain functional electric elements such as conductive tracks. Their spatial arrangement is responsible for the sensor function. In the case of the sensors investigated in this article, these are gold conductor tracks with a chromium adhesive layer between the thermoplastic substrate and the gold conductor track. In our previous work, we have already found that metallic film sensors can only be reliably microfabricated on ultra-smooth film substrates as they can be produced by spin coating [26]. It has been proven that the bond between the gold and the surrounding epoxy resin forms a weak point that can easily lead to delamination [25]. At the same time, a certain amount of gold area is necessary for the sensor function, as is clear from the example of interdigital sensors, which reach their sensitivity maximum when the electrode occupies 50% of the sensory active area [7]. In this article, we describe investigations which evaluate the potential conflict of objectives between sensor performance and the mechanical connection between the sensor and the surrounding structure. Figure 1 shows an embedding situation of a sensor for cure monitoring with some exemplary issues, highlighting the issues addressed in this article. These are, namely, the influence of the sensor structure and the substrate on the mechanical performance of the structure. In addition to mechanical performance, questions of the influence of the structure on the sensor, the resin flow interference during manufacturing, cross-sensitivity of the sensor signals or the signal transmission itself have to be addressed, when researching integrated sensors for cure monitoring but they are not subject of this work. These questions will be considered in future work and are therefore greyed out in Figure 1.

Various characteristic zones are carved out from the prototype of the sensors by abstraction and examined with the help of test specimens on carbon fibre reinforced plastic (CFRP) for the interlaminar shear strength (ILSS). The ILS specimens allow a statement to be made about the strength of the bond between the sensor structures and the surrounding matrix resin. In order to assess the crack propagation properties, dummy sensor structures with a high density of electrodes were designed and embedded into double cantilever beam (DCB) test specimens. For comparison, foils prepared by spin coating without sensor structures embedded in CFRP test specimens are also tested in both the DCB and ILS tests, as well as CFRP test specimens without integrated foils.

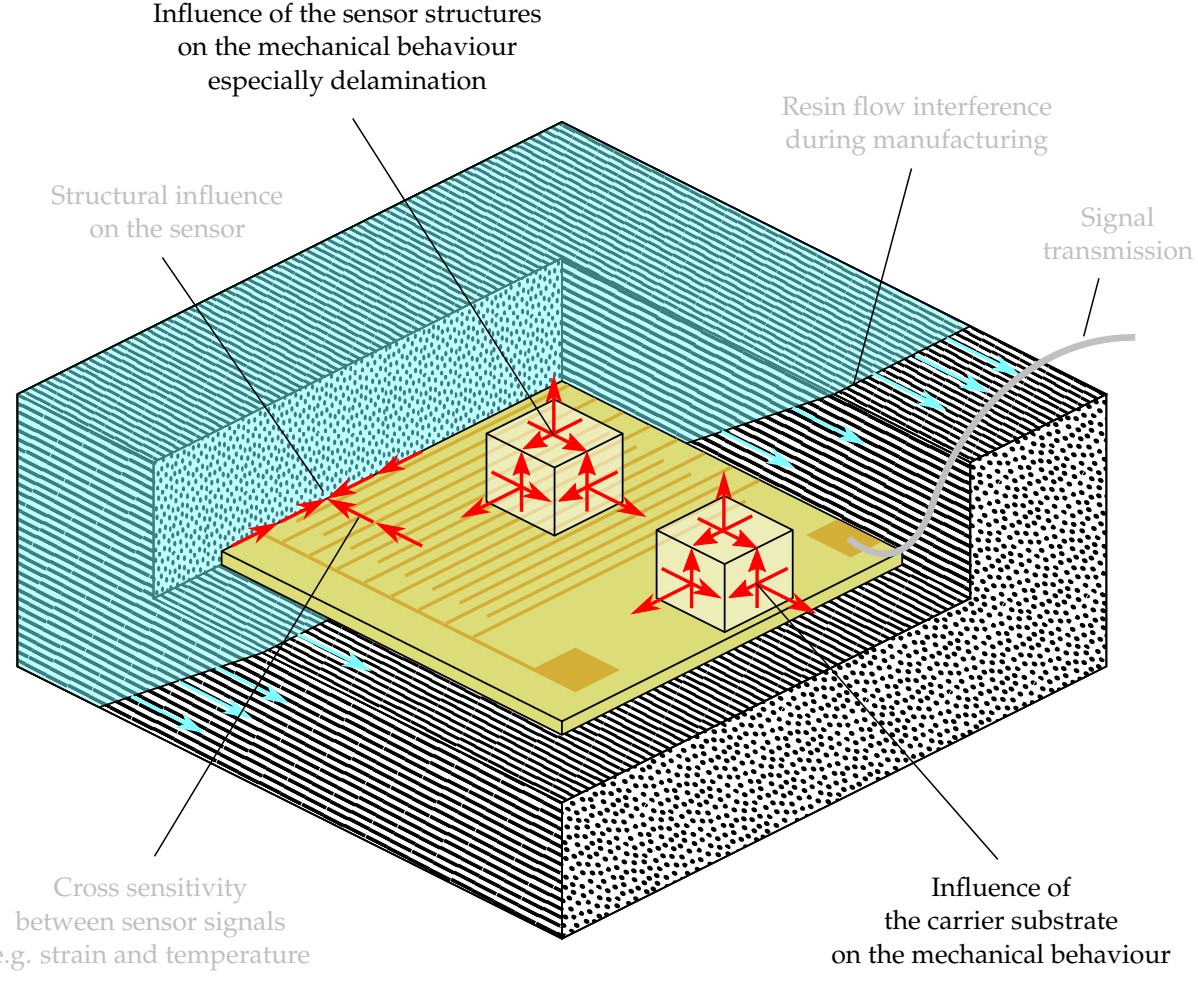

**Figure 1.** Questions arising from sensor integration into fibre-reinforced polymers.

## 2. Materials and Methods

For the production of the filigree sensor structures, fabrication processes were used as described more in detail in [27]. Figure 2a shows a sensor node designed for monitoring the progressing curing of epoxy resins. The sensor node contains a thermal sensor that eliminates strain direction sensitivity and largely suppresses the influence of the magnitude of the strain by a suitable arrangement of the conductor path [27]. In addition, the sensor node contains two strain gages in half-bridge connection with mutually orthogonal orientation and a coplanar capacitive sensor with interdigitated electrodes. Figure 2c shows the individual patterns derived from the sensor and introduced into ILS specimens. The temperature sensor structure is represented by the meandering gold conductor path, the conductor paths for the electrical connection of the sensor structures are represented by electroplated reinforced copper conductor paths and the strain gages and the interdigitated electrodes of the capacitive sensor are reproduced by conductor paths parallel to each other. As a non-isotropic behaviour is to be expected for the two line patterns, they are investigated under different angles. Figure 2b shows the cutting plan for the laser to cut the sensors out of the individual wafers.

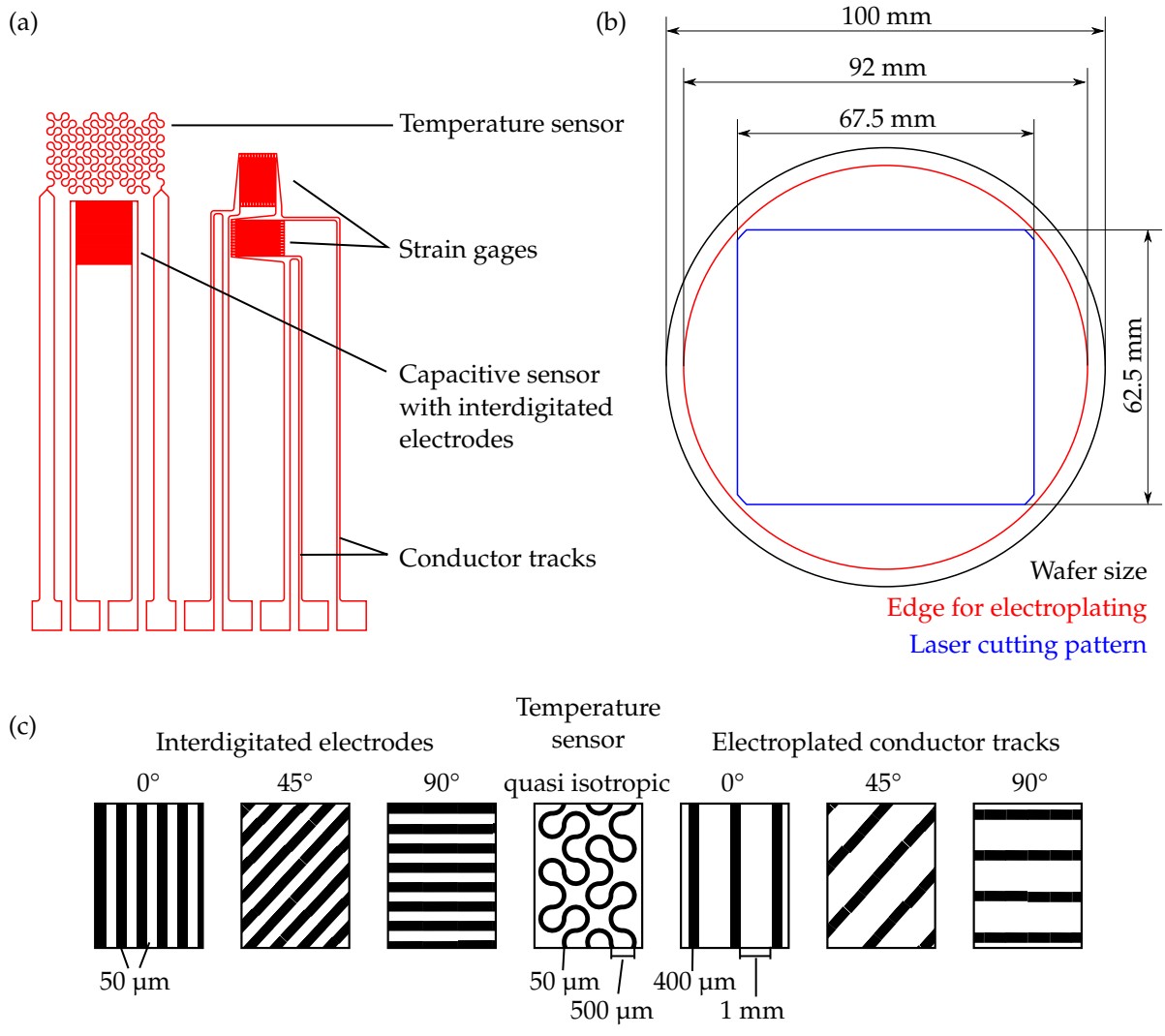

**Figure 2.** Sensor prototype and characteristic zones of the sensor geometry for ILS tests: (**a**) Sensor prototype [27]. (**b**) Laser cutting pattern. (**c**) Characteristic patterns, that were integrated into ILS specimens, not true to scale to make the conductor tracks visible.

A test pattern representing all of the typical sensor structures as shown in Figure 3 was used in the investigation of DCB specimens. During the production of the sensor samples, only the conductor paths were electroplated. Due to the relatively large test specimens, an investigation of all the test patterns shown in Figure 2 would have required the production of an unreasonably large amount of wafers. As anisotropy is to be expected due to the conductor paths, the orientations 0°, 45° and 90° were investigated mechanically.

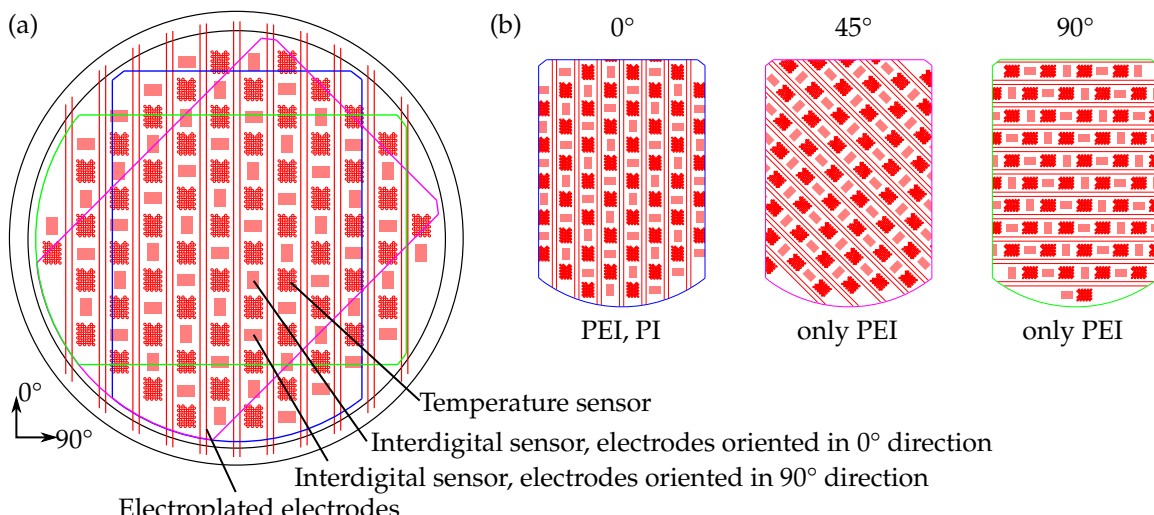

**Figure 3.** Sensor pattern for DCB testing: (**a**) Sensor pattern on a 100 mm diameter wafer. (**b**) Cut sensor patterns under different angles for integration.

Before embedding, the sensor structures were cut out using a femtosecond laser (3D Micromac, Germany, Chemnitz; 515 nm, 6 W, cutting speed 500 mm s$^{-1}$) and carefully removed from the glass wafer using a scalpel. In order to be able to handle the filigree film, it was reinforced with a TECNI Tape self adhesive film (DISCO Corp., Tokio, Japan), which was removed after application of the sensor structures to the fibre composite.

In order to investigate ILSS by means of ILS specimens, two plates with a thickness of about 2 mm were produced using the prepreg material 8552/AS4 from Hexcel (Stamford, CT, USA). Sensor structures based on PEI film were embedded in the first plate. The test specimens, which are arranged in a strip at the edge of the plate in Figure 4, were used as reference test specimens without a film being embedded in this area. Sensor structures based on PI foil were embedded in the second plate. A 50 µm thin commercial PI film (Kapton HN) from Dupont (Wilmington, DE, USA) was embedded in the specimen strip at the edge. With a nominal layer thickness of 130 µm, a laminate thickness of 2.06 mm results calculated with 16 layers, whereby the films are arranged in the centre plane, where the highest shear stresses occur in the test. Following DIN EN 2563, the laminate was constructed from UD layers, with the fibres oriented along the longest dimension of the specimens.

In order to retrieve the position of the foils during sawing, a coordinate system with vacuum foil was applied to the top of the test specimens, the edges of which are shown in blue in Figure 4. The entire assembly was evacuated and cured in a Scholz (Coesfeld, Germany) autoclave at 7 bar pressure first for 60 min at 110 °C and then for 120 min at 180 °C. The foils were provided 1.25 mm larger, so that deviations in this order of magnitude can be tolerated during the entire production. To avoid long tolerance chains, the large sheet was first sawn into several small sheets using a diamond saw from Maiko (Braunschweig, Germany), which were then cut into individual test specimens. The edges of the foils were placed on the position of the saw cuts, so that the sides of the foil, which were sometimes slightly damaged during removal with the scalpel, were removed during sawing. All saw cuts are shown as red lines in Figure 4.

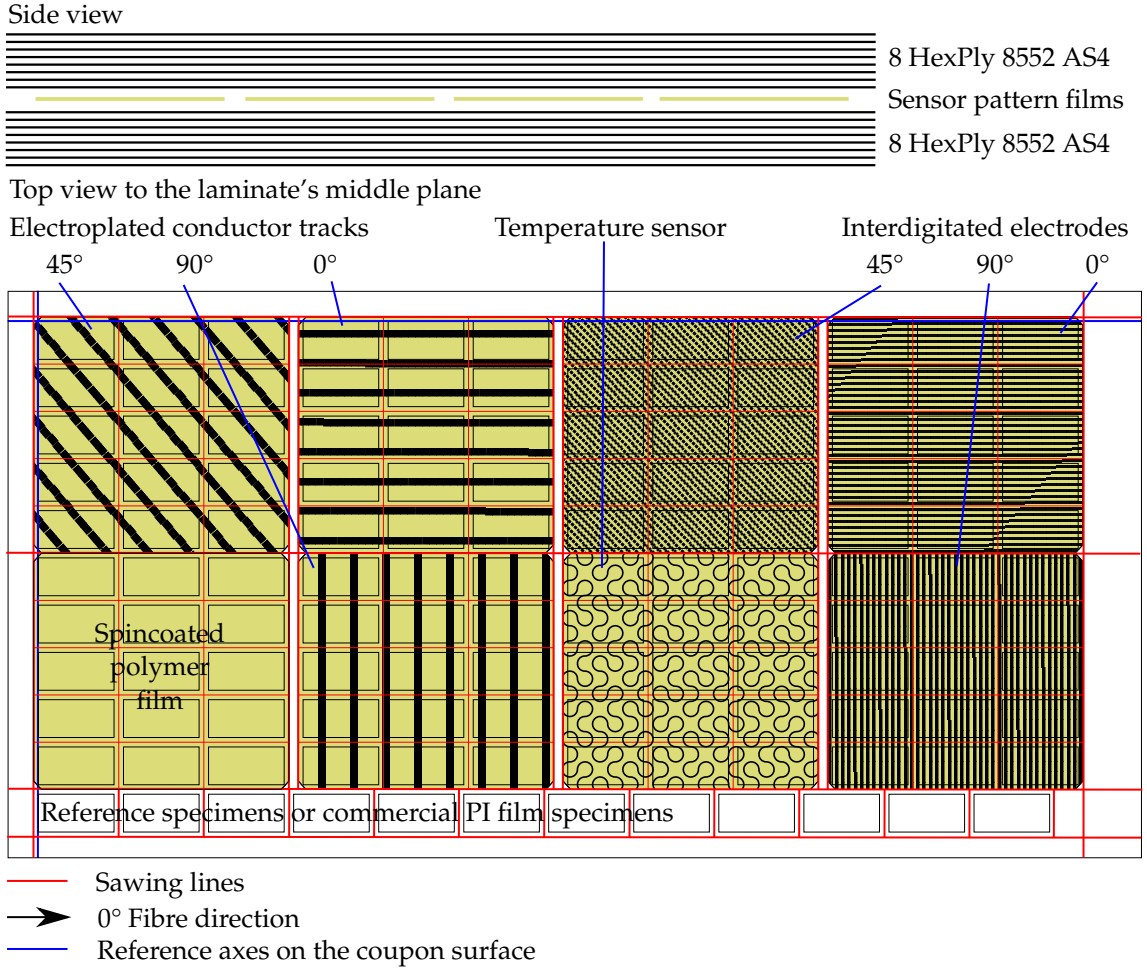

**Figure 4.** Stacking sequence, cutting plan and film placement in the ILS coupons. In the coupons with PEI films, the stripe on the side was used for reference specimens without any integrated film, and in the coupon with PI specimens, the stripe on the side was used for specimens with an integrated 50 µm thick Kapton film. The sensor patterns are shown to illustrate how the individual sensor pattern films lie in the laminate and are not to scale.

For the construction of the DCB specimens, the same Hexcel 8552/AS4 prepreg with identical curing cycle was used as for the ILS specimens. In order to achieve a laminate thickness of about 3 mm, 24 layers of prepreg were arranged unidirectionally on top of each other, with a 30 mm wide piece of release film as a precrack and the films to be tested being arranged in the centre plane between layer 12 and 13, see Figure 5. Following DIN EN 6033 and ASTM D5528, the fibres are oriented in the longitudinal direction of the specimen. As the wafer size with a diameter of 100 mm limits the length of the inserted foils, the test specimens have to be designed shorter than specified in DIN EN 6033. The hinges are also glued on the other way round than in the standard, on the one hand in order not to exceed the peel strength of the adhesive, and on the other hand to ensure comparability with the test specimens from previous investigations [25]. The adhesive used for the hinges was the 2-component epoxy adhesive 9466 from Loctite (Düsseldorf, Germany). Before gluing, the hinges were sanded with fine sandpaper and thoroughly cleaned with isopropanol.

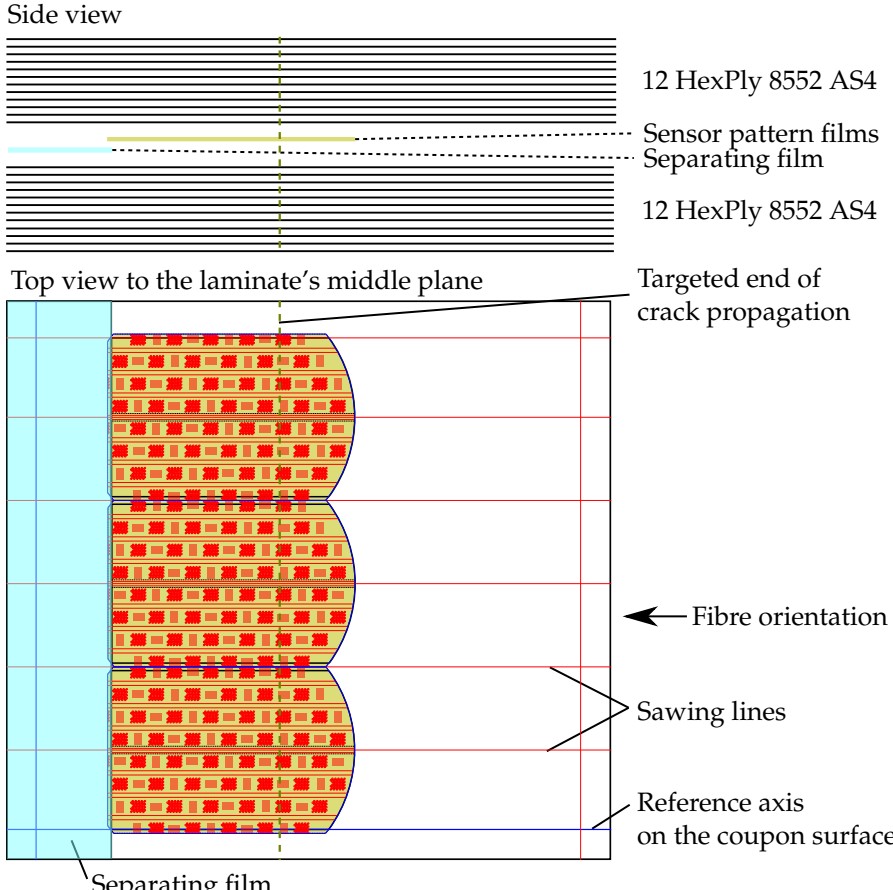

Side view

12 HexPly 8552 AS4

Sensor pattern films
Separating film

12 HexPly 8552 AS4

Top view to the laminate's middle plane

Targeted end of
crack propagation

Fibre orientation

Sawing lines

Reference axis
on the coupon surface

Separating film

**Figure 5.** Stacking sequence, cutting plan and film placement in the DCB coupons. A coupon containing a sensor pattern in 0° direction is shown. For each orientation of the sensor pattern a laminate as displayed was manufactured. Coupons were manufactured with the following integrated films: PEI @ 0°, 45° and 90°, PEI without sensor pattern, PI @ 0° and without sensor pattern. As a reference, the experimental results from an earlier investigation [25] made from the same material (HexPly 8552 AS4) were used. The green dashed line indicates the final crack length pursued in the DCB tests.

An Instron (Norwood, MA, USA) column testing machine with a 30 kN load cell was used to test the specimens, furthermore a test device for 3-point bending, which has a support spacing of 10 mm and radii of 3 mm following DIN EN 2563. The interlaminar shear strength $\tau_{ILSS}$ results from the force at failure of the specimen $F_B$ by the relation

$$\tau_{ILSS} = \frac{3 \cdot F_B}{4 \cdot b \cdot h} \tag{1}$$

where $b$ and $h$ indicate width and thickness of the specimen.

A Zwick/Roell (Ulm, Germany) universal testing machine was used to perform the DCB tests. Like the ILS specimens, the DCB specimens were not subjected to any special preconditioning. In Figure 5, it can be seen that each laminate allows the production of six specimens. Of these six test specimens, one was used to set up the testing machine and the other five test specimens were used to determine the critical energy release rate. The setting of the testing machine includes the definition of a holding phase to mark the crack length on both sides of the specimen and the displacement of the crosshead at which the machine should stop crack propagation to mark the final crack length on both sides of the specimen. For the holding phase, the testing machine was set for each series to achieve a crack length of about 30 mm counted from the hinge of the specimen. Due to the strongly different crack propagation resistances of the specimens with integrated PEI film and PI film, the displacements in the holding phase also differ considerably between the specimen

series (1.5 mm for specimens with PI film versus 3.7 mm for specimens with PEI film). The displacement at the stop of the testing machine was set to a desired final crack length of about 80 mm so that the crack does not leave the area of the inserted film. Again, the displacements differ considerably due to the different crack propagation resistances (6 mm for PI film versus 18.7 mm for PEI film). This circumstance can also be seen later on when presenting the results.

### 3. Results

Figure 6 shows the course of the interlaminar shear stress $\tau_{ILSS}$ calculated from the compressive force *F* and the specimen geometry plotted over the displacement. Figure 6a shows the curves of the reference specimens, and Figure 6b the curves for the specimens with a spin-coated PEI film without sensor structures, which seem to be indistinguishable from those in the reference specimens. Figure 6c shows the course of the interlaminar shear stress for test specimens with an embedded spin-coated PI film. Here, a very early, clear drop in force can be seen after which the force–traverse path curves continue to run with a reduced slope. This drop in force, which is accompanied by an increase in compliance, was interpreted as failure of the test specimens. Figure 6d finally shows the course of the interlaminar shear stress for the test specimens with PEI film with thin conductive tracks in 0° orientation to the fibre direction. A much smoother force collapse than in Figure 6c can be seen just above 2 kN. Above this force breakdown, the curves continue with reduced slope, indicating an increase in compliance.

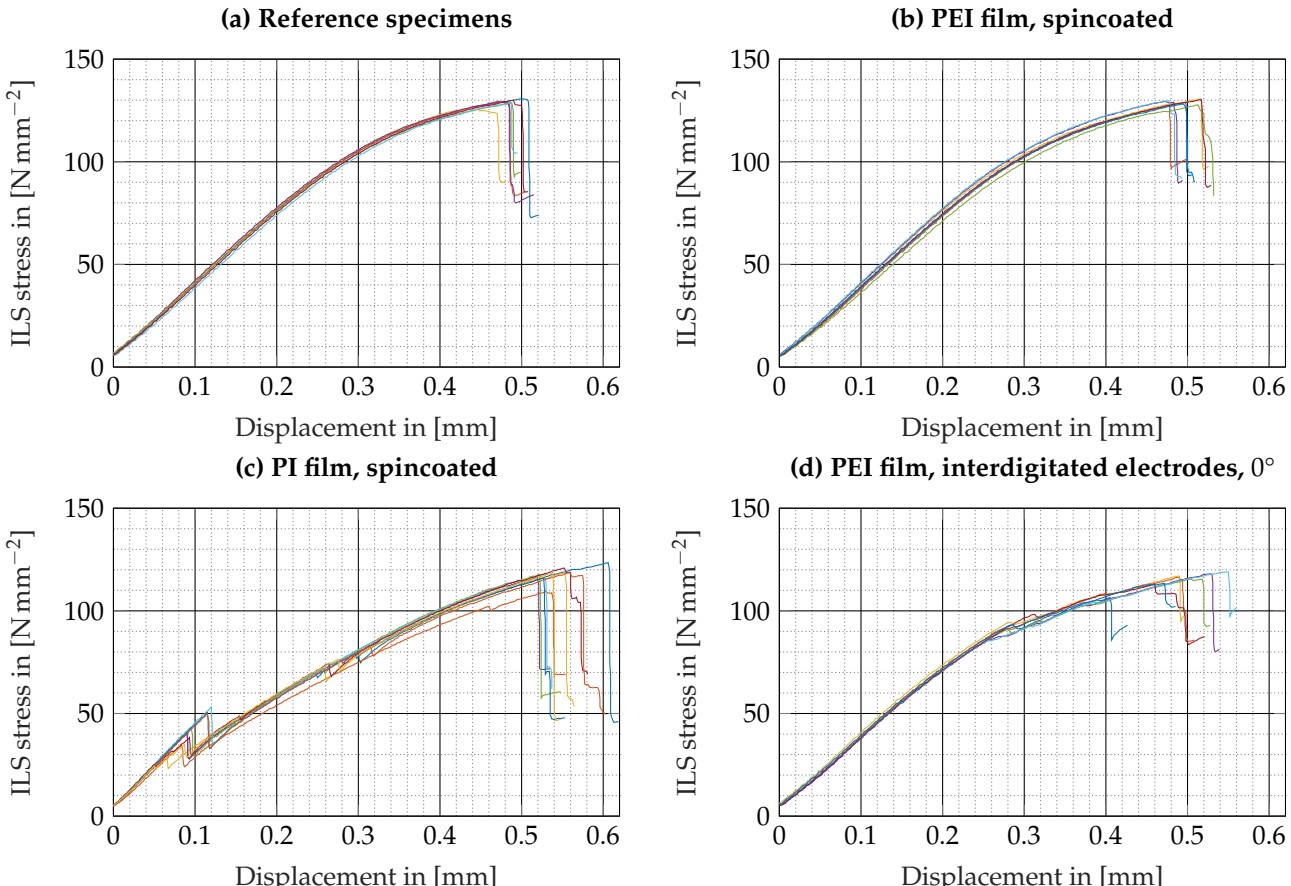

**Figure 6.** Bending force over displacement curves for four types of failure in ILS specimens: (**a**) Reference specimens. (**b**) PEI film, spincoated, without sensor structures. (**c**) PI film, spin coated, without sensor structures. (**d**) PEI film, spin coated, with interdigitated electrodes, orientated in fibre direction.

Table 1 shows which specimens showed which stress deflection characteristic during the test. For the specimens with thin tracks at angles 45° and 90°, clear categorisation was not possible, because on the one hand these specimens did not show any load drop before reaching their maximum strength, which is why the curves show similarity to Figure 6b. On the other hand, in the differential compliance curves in Figure 7, a clear increase in compliance can be seen well before the specimen fails, unlike in the specimens with spin-coated PEI film without sensor structures. However, as no clear force collapse could be seen, the value of the force at the first force collapse was chosen as the ILSS for the calculation of Figure 8. This determination must be taken into account when interpreting Figure 8. A closer look at the compliance curves is therefore worthwhile.

**Table 1.** Damaging modes compared to the modes shown in Figure 6. The "*" symbol indicates the specimen groups, that belong to the diagrams shown in Figure 6b–d.

|  | PEI | PI |
|---|---|---|
| Commercial film | b | b |
| Spincoated | b * | c * |
| Temperature sensor | b | c |
| Thin tracks @ 0° | d * | c |
| Thin tracks @ 45° | b / d | c |
| Thin tracks @ 90° | b / d | c |
| Electroplated track @ 0° | b | c |
| Electroplated track @ 45° | b | c |
| Electroplated track @ 90° | b | c |

For the calculation of the interlaminar shear strength and the representation in Figure 8, the load value at the first force breakdown of the displacement force curve was applied, as specified in DIN EN 2563. The bar chart shows the mean values of the individual specimen series and the error bars mark the 95% confidence intervals from the one-sample *t*-test in each case. At least seven specimens were tested in each specimen series. Figure 8 shows that the fine interdigitated electrodes affect ILSS even when using a PEI substrate, while all other tested sensor structures and also the PEI film without sensor structures have no influence on ILSS. Comparison with studies on PEI films fully coated with gold [25] suggests that the gold content in the area has a significant influence on the weakening of the connection, whereby the zones between the interdigitated electrodes with epoxy-PEI connection seem to exert a stabilising effect. While the commercial PI film also does not exert a significant influence on ILSS, all films of spin-coated PI show a dramatic drop in interlaminar shear strength. Due to the existing scatter, most of the differences between the different sensor patterns cannot be identified as significant. The strength value of the specimens with interdigitated sensor pattern is noteworthy, showing both the lowest mean value and the lowest scatter of all specimens with PI film. The impairment of the bond appears to be so big due to the PI film alone that the sensor patterns play a subordinate role in the weakening. In contrast, foils with electroplated copper tracks even seem to have a stronger connection to the rest of the composite, possibly due to effects on the PI surface during the electroplating process.

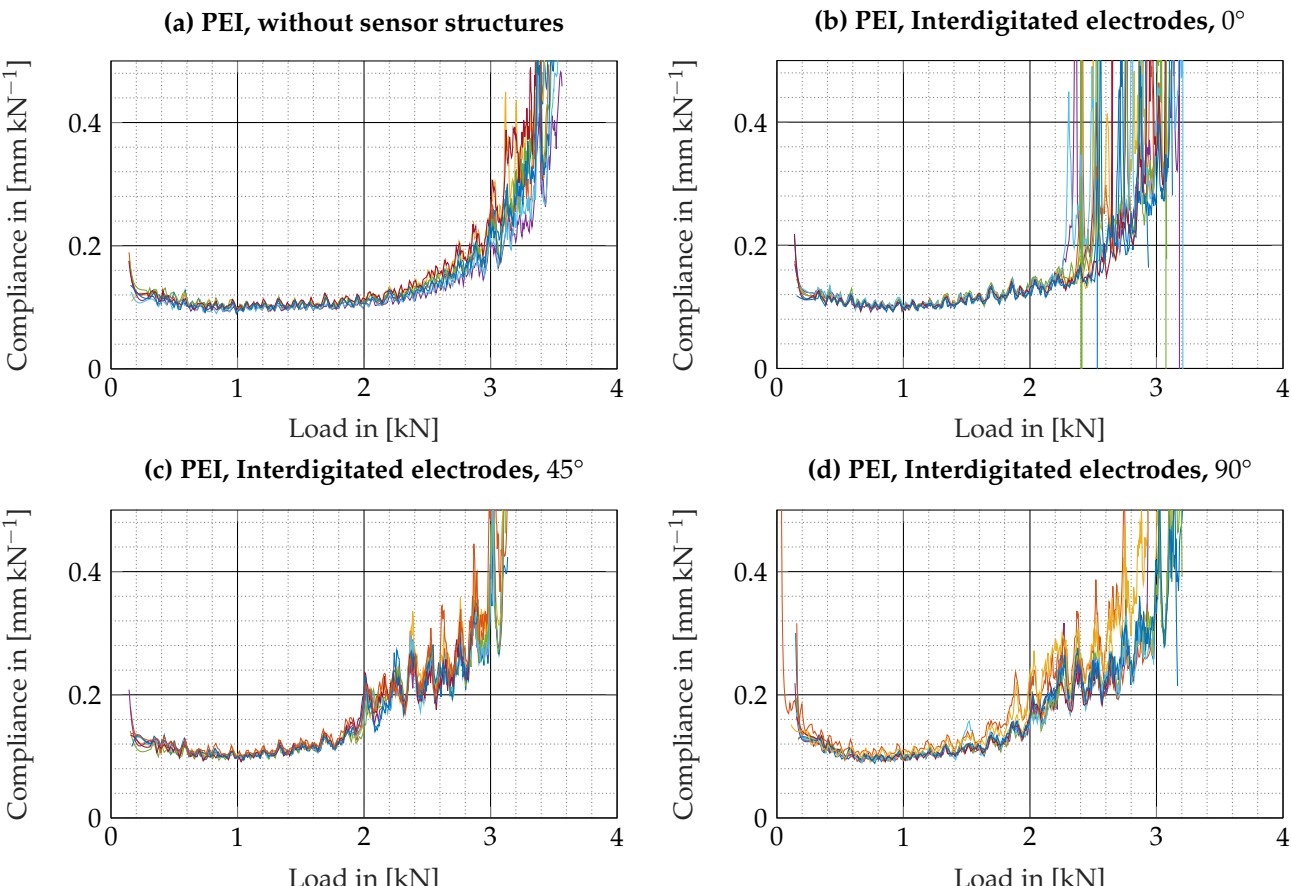

**Figure 7.** Compliance over bending force curves for several ILS specimens with spin-coated PEI film: (**a**) PEI film, without sensor structures. (**b**) PEI film, with thin tracks @ 0°. (**c**) PEI film, with thin tracks @ 45°. (**d**) PEI film, with thin tracks @ 90°.

Figure 7 shows the differential compliance curves of the specimens with spin-coated PEI film without sensor structures as well as with interdigitated electrodes at the angles 0°, 45° and 90° over the compressive force $F$. The differential compliance $\delta$ results from the derivation of the deflection $x$ according to $F$ with the right-sided difference quotient. A smoothing of the deflection and the force curve over 25 data points each suppresses strong noise in the calculated difference quotient. Since 25 data points make up between 5 and 10% of the data vector, a falsification of the original curve must be avoided unconditionally. For this purpose, the Savitzky–Golay method with a fifth-degree interpolation polynomial is used for smoothing. The interpolation polynomial also correctly represents the load drops, which can be seen in Figure 7b by negative compliance values for compressive forces above 2.4 kN. The curves in Figure 7b–d show that the compliance in the range from approximately 2.0 to 2.5 kN is significantly increased compared to the pure PEI film without sensor structures. Thereby, negative compliance values only occur in the specimens with interdigitated electrodes at an angle of 0°, which fits with the fact that early load collapses were only found for these specimens.

Furthermore, for the DCB test specimens, it is worth looking at the load–displacement curves first. Figure 9a shows the load-displacement curves of the specimens with spin-coated PEI film without sensor pattern, representative for all specimens with PEI film. In all load–displacement curves of specimens with PEI film, a clear crack jumping can be observed, recognisable by the sudden drops in force and subsequent slow increases in force. Videos taken during the test also confirm this behaviour. In the specimens with PI foil in Figure 9b it can be seen that all curves are at a significantly lower force level and already at a much lower displacement the final crack length was reached and the test stopped.

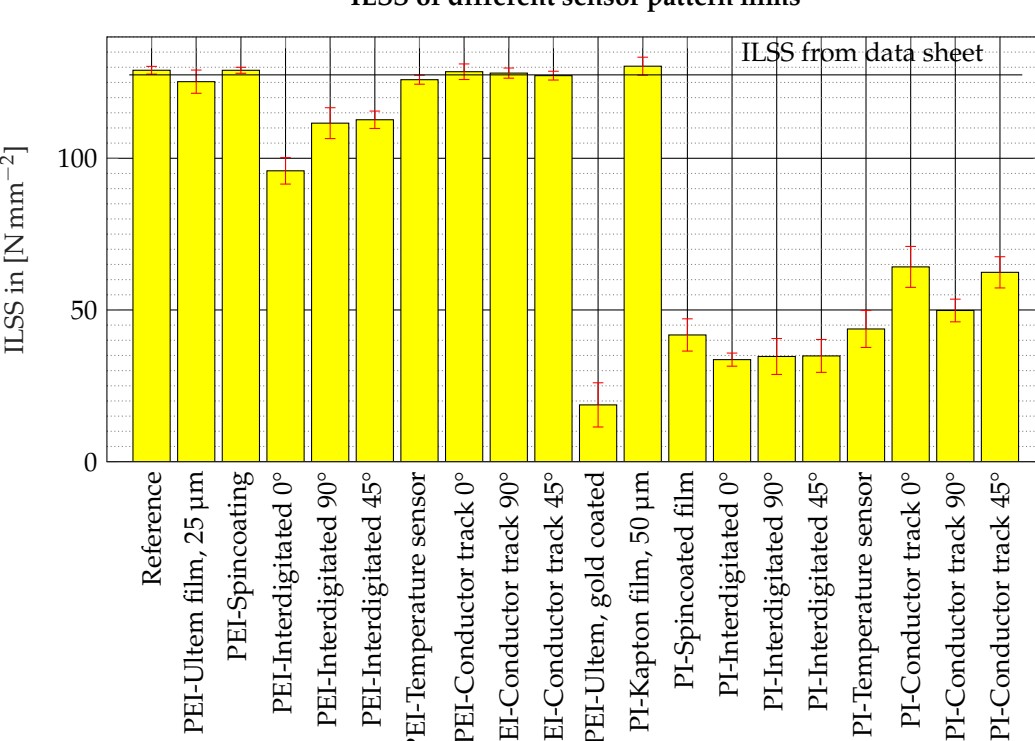

**Figure 8.** Comparison of the ILSS of different specimen series. The data on the ILSS of specimens with integrated 25 μm Ultem film (PEI) and gold-coated Ultem film comes from [25].

The significantly lower load level and the significantly lower displacements of the crossheads when reaching the final crack length are also reflected in the calculated critical energy release rates, which are shown in Figure 10. The bars indicate the mean values of the respective specimen series and the error bars mark the 95% confidence intervals from the one-sample *t*-test. At least five specimens were examined per specimen series. For comparison, the bar chart also shows the critical energy release rates of reference specimens and specimens with integrated commercial PEI film (25 μm thick Ultem film, Goodfellow GmbH, Hamburg, Germany) and PI film (50 μm thick Kapton film, Dupont, Wilmington, DE, USA) manufactured using the same HexPly8552 AS4 prepreg material [25]. The diagram clearly shows that both the commercial PEI film and the spin-coated PEI films result in a significant increase in the critical energy release rate. The application of the sensor patterns also has no significant negative effect on this increase, although the measured mean values are somewhat lower than the mean value of the test specimen series with spin-coated PEI film without sensor patterns. However, due to the scattering, which is certainly also due to the observed crack jumping, no significant negative influence of the sensor patterns can be proven with the available data. An angle dependency cannot be proven on the basis of the available data either. Compared to the reference, the PEI films with sensor patterns also show a significant improvement in the critical energy release rate. For the commercial PI film, no influence can be demonstrated compared to the reference [25], but the spin-coated PI film reduces the critical energy release rate significantly and also statistically significantly by 85%. Compared to the spin-coated PI film, the sensor structures even increase the critical energy release rate from 34 kJ m$^{-2}$ to 87 kJ m$^{-2}$.

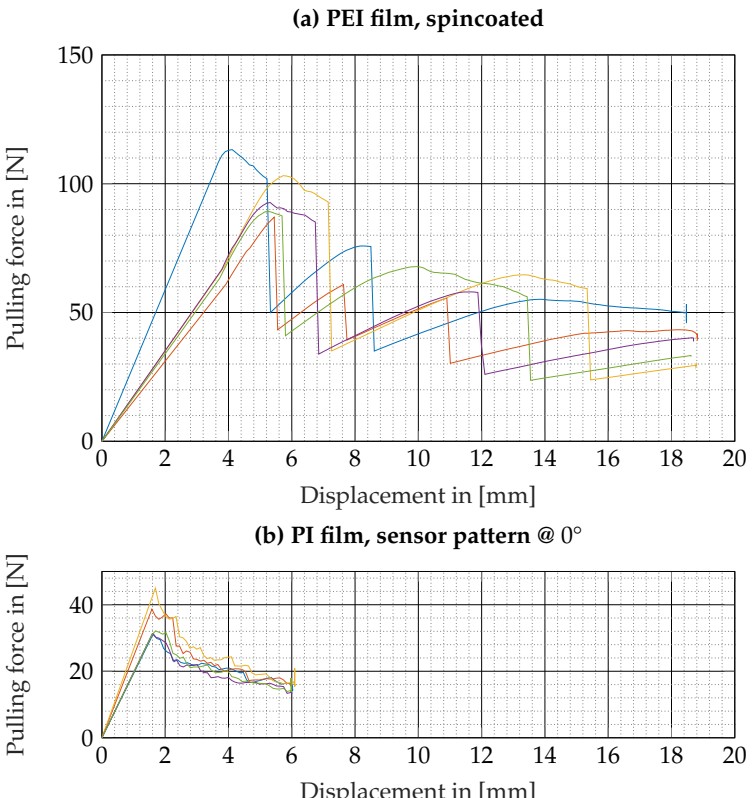

**Figure 9.** Force displacement curves for DCB specimens with spin-coated integrated films: (**a**) Pronounced crack jumping visible as force breakdowns in the specimens with integrated spin-coated PEI film. That behaviour is visible in all specimens with spin-coated PEI film with and without sensor patterns. (**b**) No crack jumping in specimens with PI film and applied sensor pattern @ 0°.

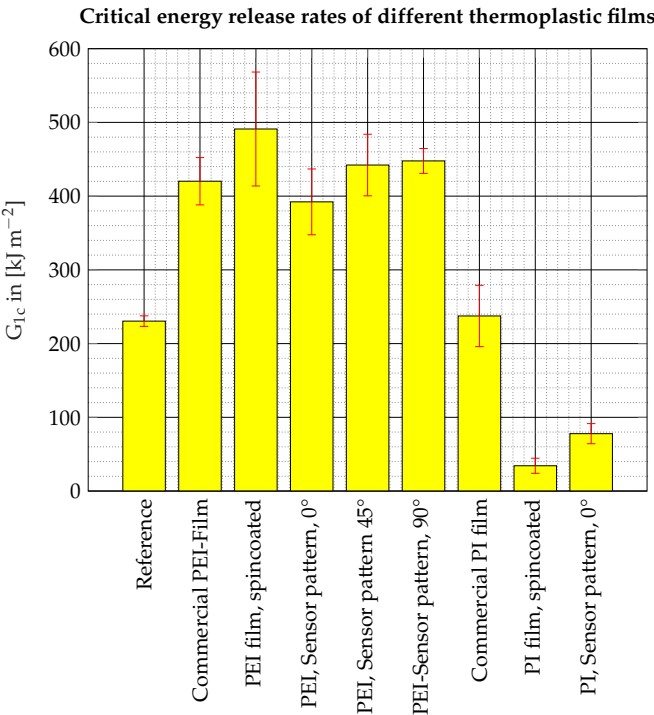

**Figure 10.** Critical energy release rates from DCB tests for different film materials and different orientations of a universal sensor structure. The data on the reference specimens and the commercial PEI and PI film is from [25].

## 4. Discussion

The results of the ILS examinations allow four conclusions to be drawn:

- ILSS is negatively affected by the gold structures at the interface.
- The thickening of the connecting conductor tracks due to electroplating has no decisive influence on ILSS.
- The angle between the interdigitated electrodes and the occurring shear stress influences the measured ILSS.
- Commercial PI film and spin-coated PI film behave substantially differently in the composite.

The four conclusions can be substantiated not only by the bar charts and compliance curves in Figures 7 and 8, but also by fracture images of the ILS specimens taken with a light microscope (Smartzoom 5, Zeiss, Germany, Oberkochen). In each case, specimens were selected that were representative of the respective specimen series, as the fracture images of all specimens cannot be shown for reasons of limited space. Figure 11a shows a fracture image of a reference specimen for comparison, where it is visible that due to the parabolic shear stress profile, the cracks run close to the mid-plane, but rarely exactly in the mid-plane. The fracture images of specimens with spin-coated PEI film show that the cracks are not in the mid-plane, which at least proves that the spin-coated film does not weaken the specimen compared to the base material. The same behaviour can be observed for the test specimens with electroplated copper tracks (Figure 11c) and for the test specimens with commercial PI film (Kapton). In Figure 11c, the mid-plane is particularly visible due to the clearly visible copper tracks, and in Figure 11g, the 50 µm thick commercial PI film shows up as a dark orange line. The crack images match that no change in shear strength can be seen in the bar graph in Figure 8 because the cracks occurred away from the bond, so ILSS of the base material HexPly 8552/AS4 is decisive and the measured interlaminar shear strengths can only be a lower limit for the true shear strength of the contact zone. Figure 11d–f shows a separation of the laminates in the mid-plane, in many specimens with spin-coated PI films with and without sensor structures even a complete separation of the two halves of the specimen was observed, so that the film became visible again on the surface of one of the two halves. The fracture images match the lower strength values for all spin-coated PI films in Figure 8 and show that the respective embedded foils resulted in a local weakening in the mid-plane.

The measured configurations clearly show that ILSS is not linearly related to the gold content in the surface. If no gold is applied to the PEI film, the cracks do not run at the interface between PEI and epoxy, but in the base material, so that the shear strength of the base material is decisive for the value measured in the test, see Figure 11b and the work in [25]. For the temperature sensor structure with a low gold content, this behaviour remains the same; however, for higher gold contents of 50% and more, the bond between the sensor foil and the epoxy resin becomes the weak point in the ILS test. While a gold area coverage of 50% by interdigitated electrodes represents an optimum with regard to sensitivity [7], from a mechanical point of view, a reduction of the area infill at the expense of sensitivity seems reasonable in order not to impair ILSS. Accordingly, the sensitivity and the degradation of the interlaminar shear strength represent opposing receivables.

Comparing different angles of the interdigitated electrodes shows that not only the areal fraction of the gold, but also the geometrical arrangement has a significant influence on the damage. If the conductors are aligned in the direction of the shear stress and fibre direction, a lower ILSS results in the test. However, the observation of the compliance curves clearly shows that a degradation of the compliance also occurs in the test specimens with conductive tracks under 45° and 90° orientation to the fibre direction and this well before the first force breakdown. A conceivable explanation would be plastic flow of the introduced gold before the actual failure of the composite. The orientation of the interdigitated electrodes therefore does not change the fact that the local load-bearing capacity of the composite is impaired, but it does have an influence on the damage behaviour during the test.

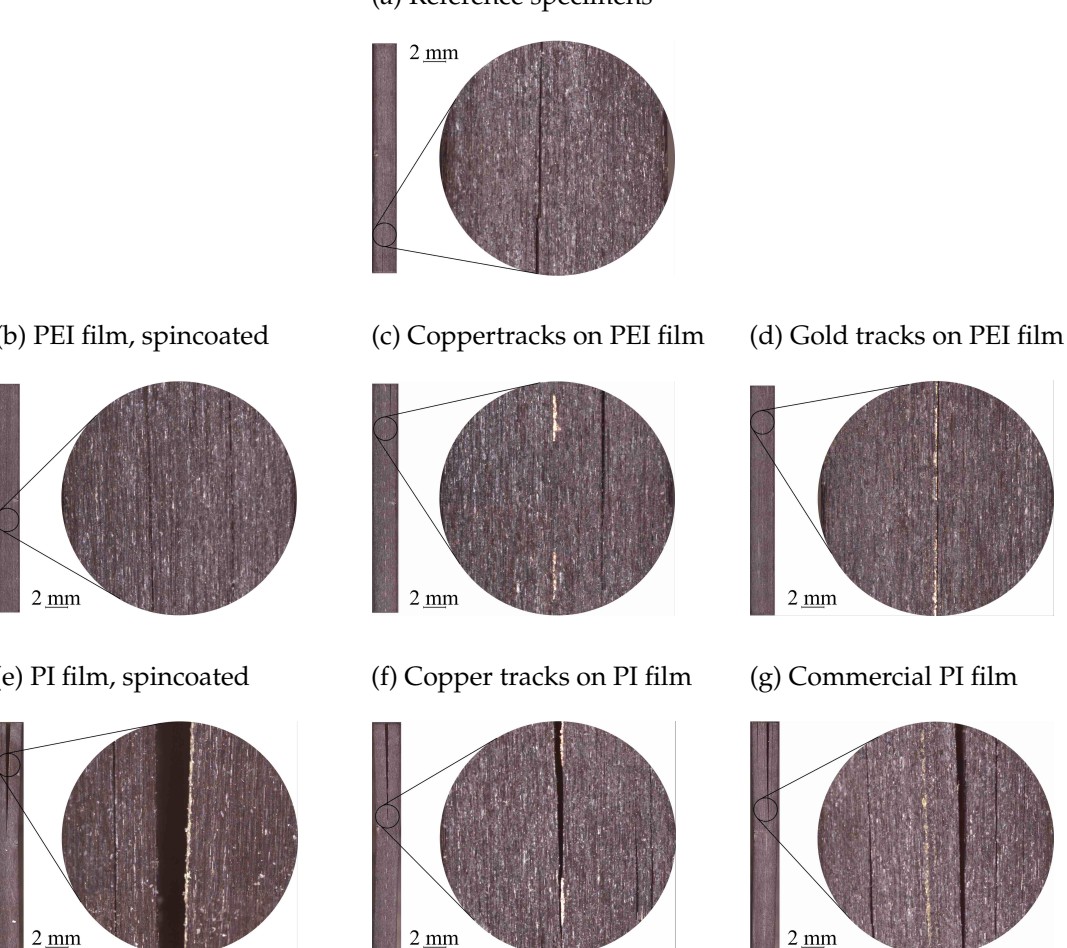

**Figure 11.** Side view of fractured ILS specimens: (**a**) Reference specimens. (**b**) PEI film, spin coated, without sensor pattern. (**c**) PEI film, spin coated, with copper tracks @ 90°. (**d**) PEI film, spin coated, with thin gold tracks @ 0°. (**e**) PI film, spin coated, without sensor pattern. (**f**) PI film, spin coated, with copper tracks @ 90°. (**g**) Commerial PI film (Kapton).

While very densely arranged thin conductive tracks of the interdigitated electrodes impair ILSS, the investigations show that widely spaced conductive tracks but instead thickened by electroplating have no negative influence on ILSS. The strong bonding of the foil [24] seems to compensate for the negative effects of the additionally introduced material. That a considerably better bonding of copper to epoxy resin could be responsible for this observation seems unlikely against the background of earlier investigations, in which a detachment of the chromium adhesion layer between the thermoplastic film and the epoxy resin was also observed in some cases [25]. Since this weak point is connected in series with the interphase between copper and epoxy resin, strong adhesion between copper and epoxy resin would probably only cause the chromium adhesion layer to detach from the thermoplastic film instead. The supporting effect of adjacent epoxy–PEI interfaces is therefore the more likely explanation. A similar supporting effect has already been discussed as an argument for the miniaturisation of embedded PI film sensors elsewhere [16].

However, the decisive influence on ILSS is ultimately not the sensor structure itself, but the choice of thermoplastic substrate film. Interestingly, there is even a difference between films of the same material, because while the commercial PI film exerts no influence on ILSS, ILSS drops dramatically for the spin-coated PI film. One conceivable explanation for this is the very smooth surface of the spin-coated PI film, which does not allow any mechanical snagging in surface roughness. In contrast, the commercial PEI film and the spin-coated PEI film do not exhibit any difference regarding their influence on ILSS. The

results of the DCB tests fit into the picture obtained from the ILS tests. Both the commercial and spin-coated PEI films improve the critical energy release rate to about double. It is also clear from Figure 12a,b that the PEI film is no longer removable on the surface and has formed a strong bond with the surrounding composite during fabrication. The different texturing of the individual crack zones, which is particularly evident in Figure 12a, also matches the crack jumping observed for the test specimens with PEI films as shown in Figure 9a. The sensory patterns do not exert a decisive negative influence, as the critical energy release rate is higher than for the reference laminate despite the sensory structures. While the commercial PI film, in addition to ILSS, does not significantly affect the critical energy release rate, the spin-coated PI film reduces it by 85% from the reference value of 230 kJ m$^{-2}$ to 34 kJ m$^{-2}$. The adhesion between the PI film and the surrounding composite is so weak that the film can be peeled off again, as indicated in Figure 12c. The sensor patterns even exert a positive influence on the PI substrate, increasing the critical energy release rate from 34 kJ m$^{-2}$ to 87 kJ m$^{-2}$. One explanation for this is that surface changes in the PI film due to the electroplating process are responsible for the increase in the critical energy release rate. A possible influence of the electroplating process was already discussed for the copper tracks in the ILS specimens. Another possible explanation is that plastic deformations in the metallic sensor structures consume additional energy. Which of the two influences dominates cannot be decided on the basis of the available data.

The fracture images in Figure 12b,d show sensor structures on both sides of the crack for both the PEI film and the PI film. In some areas in Figure 12b no sensor structures can be seen on either side of the crack which can be explained by crack deflection into the laminate. In Figure 12d, some golden areas can be seen on the lower half of the specimen and coppery areas on the upper half of the specimen, furthermore the thermoplastic film is clearly visible on the upper part of the specimen. At positions where sensor structures are visible on the top side, no sensor structures are visible on the bottom side, and vice versa. The most probable explanation is that predominantly the respective metal–polymer interfaces have broken. Golden areas indicate, that the gold–epoxy interface has failed and the coppery areas indicate a failure of the interface between the chromium adhesion layer and the thermoplastic film since viewing the gold layer through the 10 nm thin chromium layer the sensor structure appears coppery. The crack has thus propagated through the sensors, indicating that the measured critical energy release rate is indeed relevant to the interface between the sensor structure and the surrounding composite. At the same time, the applied sensor structures introduce a weak point regarding crack propagation into the fibre composites. However, a carrier material with a sufficiently strong bond to the composite, such as PEI, can also exert a supporting effect with regard to the critical energy release rate and more than compensate for the negative influence of the sensor structures.

(a) PEI film, spincoated

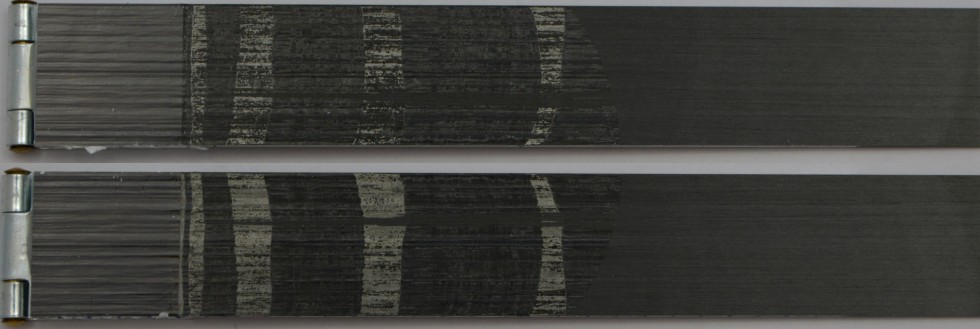

(b) PEI film with universal sensor pattern, 0°

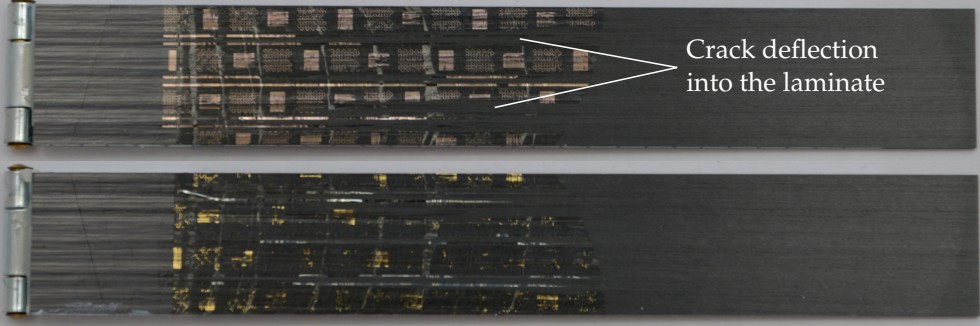

Crack deflection into the laminate

(c) PI film, spincoated

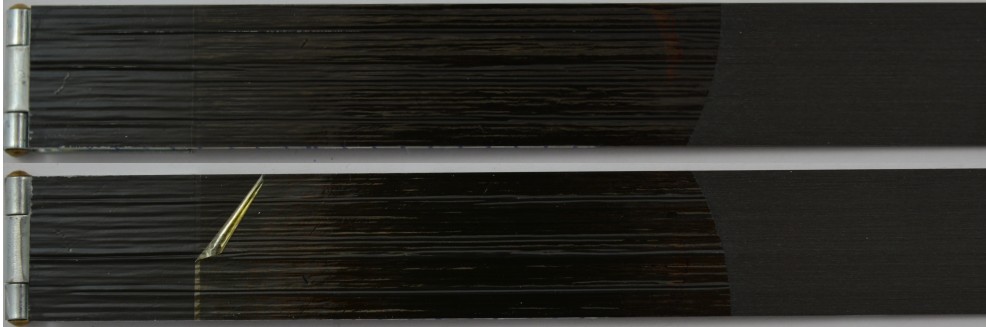

(d) PI film with universal sensor pattern, 0°

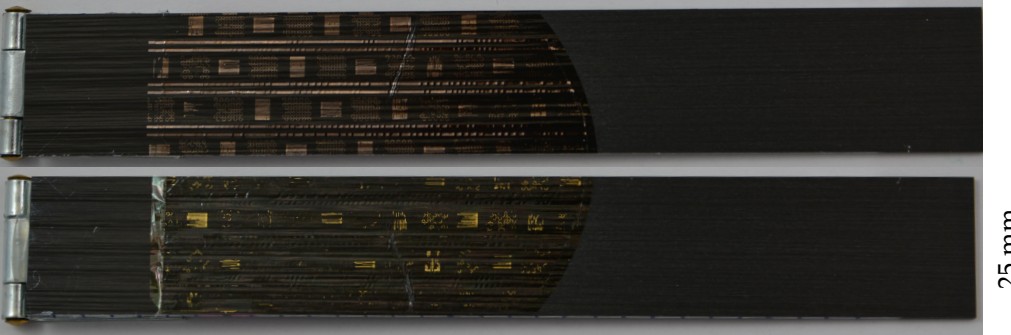

25 mm

**Figure 12.** Fracture surfaces of DCB specimens: (**a**) PEI film, spin coated, without sensor pattern. (**b**) PEI film, with 0° sensor pattern. (**c**) PI film, spin coated, without sensor pattern. (**d**) PI film, spin coated, with 0° sensor pattern.

## 5. Conclusions

The test results presented in this article show that sensor structures made of gold can negatively influence the failure behaviour of fibre-reinforced epoxy resin. ILSS is decreased down to 11% of the reference specimen strength for composites with integrated fully gold coated PEI films [24]. As the sensor structures are introduced into the composite on a carrier

material, the negative influence of the sensor structures can be more than compensated by smart choice of the carrier material. For sensor structures on PEI with an area share below 30% no deterioration of ILSS is observed at all. The reason for this seems to be a supporting effect of interfaces between the carrier material and the epoxy resin, which are adjacent to weak interfaces between gold and the epoxy resin. Such a support effect also underlines the advantageousness of miniaturisation for embedded foil sensors in reducing the 'wound effect' and the positive effect of providing holes in PI carrier films [16].

The supporting effect finds its limits when the sensor structures make up a large proportion of the contact area. This applies, for example, to interdigitated electrodes, where the highest sensitivity is achieved for a gold content of 50% [7]. For that area share of gold, a reduction in ILSS of about 26% is observed. On the other hand, the design of the interdigitated electrodes could also be adapted to a design with a lower gold content at the expense of sensitivity per area. The finger-shaped electrodes would have to be made thinner than the gaps between them.

The results also show that in the case of sensors on PEI substrates, miniaturisation seems less important for reducing degradation and larger areas become potentially attainable. Thus, higher sensitivities can be achieved and the sensors become potentially accessible for simpler readout electronics and more disturbance-intensive environments. Furthermore, this work shows that PEI foil preparation by spin-coating of a liquid precursor produces films with sufficiently smooth surface for microsensor fabrication and with high interfacial stability when integrated in epoxy based composite materials. This makes PEI one of the most promising materials for sensor integration into fibre-reinforced plastics, as it shares welcome properties such as high strength, low stiffness jump and sufficient flexibility with PI, while still allowing stronger adhesion.

**Author Contributions:** Conceptualisation, A.K. and K.R.; methodology, A.K., K.R. and C.v.d.H.; software, A.K. and K.R.; validation, A.K.; formal analysis, A.K. and J.F.; investigation, A.K. and J.F.; resources, K.R.; data curation, A.K. and J.F.; writing—original draft preparation, A.K.; writing—review and editing, A.K., J.F., K.R. and C.v.d.H.; visualisation, A.K. and K.R.; supervision, M.S. and A.D.; project administration, M.S. and A.D.; funding acquisition, M.S. and A.D. All authors have read and agreed to the published version of the manuscript.

**Funding:** This research was funded by the German Research Foundation (DFG, Deutsche Forschungsgemeinschaft), grant number 397053684, "Eingebettete multifunktionale Sensoren zur Steuerung des Aushärteprozesses von Faserverbunden". We acknowledge support by the German Research Foundation and the Open Access Publication Funds of Technische Universität Braunschweig.

**Institutional Review Board Statement:** Not applicable.

**Informed Consent Statement:** Not applicable.

**Data Availability Statement:** The test data generated during the experiments can be requested from the authors.

**Acknowledgments:** We acknowledge support by Jan Niklas Haus from IMT for laser cutting of the sensor foils.

**Conflicts of Interest:** The authors declare no conflicts of interest. The funders had no role in the design of the study; in the collection, analyses or interpretation of data; in the writing of the manuscript; or in the decision to publish the results.

## Abbreviations

The following abbreviations are used in this manuscript:

| | |
|---|---|
| CFRP | Carbon Fibre-reinforced polymer |
| DCB | Double cantilever beam |
| ILS | Interlaminar shear |
| ILSS | Interlaminar shear strength |
| PEI | Polyetherimide |
| PI | Polyimide |

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
