# Peer review of "Reducing the Weakening Effect in Fibre-Reinforced Polymers Caused by Integrated Film Sensors"

_jcs, doi:10.3390/jcs5100256_

Round 1

Reviewer 1 Report

The article is written very well. The work deserves special recognition. The substantive contribution of the paper is at a very high level. The authors have ensured that the readers can enjoy reading this paper. The subject of study is presented in a very thorough manner, and the discussion section provides a significant amount of relevant information. The paper deserves to be published as submitted to the journal.

Author Response

Dear Reviewer,

thank you very much for the positive feedback on my article.

Kind regards,
Alexander Kyriazis

Reviewer 2 Report

This manuscript treats an important and interesting topic for the relevant scientific field. The manuscript is well constructed and introduction contains enough reference. In addition, the novelty of the study and the methods are explained in the manuscript. Results and discussions are presenting clearly the obtained results. To sum up, this manuscript is recommended to be published in the journal after correcting a few typing mistakes and enriching the discussion with the references.

  • Line 41 : grammatical mistake
  • Line 133 : More in detail...
  • Line 150-231-237-245-286-332-333 'F' igure
  • Line 238 "that" is useless.- Line 322 "Figure" word is missing.
  • More referencing can be done in the discussion in order to strengthen the arguments proposed. 

Author Response

Dear Reviewer,

thank you very much for the positive and constructive feedback on my article.
I integrated the proposed improvements into the article.
A point-by-point answer is in the attachment.

Kind regards,
Alexander Kyriazis

Reviewer 3 Report

Integrating sensors in composites can be used to monitor the distribution of temperature field in the forming process of composites. At the same time, the integrating of sensors will weaken the mechanical properties of composites. This paper 's choice is interesting. This paper investigated the influence of the type and structure of the sensor film substrate in the composite material and the electrode arrangement of sensor on the bending and shearing properties of the composite material. Simultaneous interpreting the influence of the different sensor matrix on the interface bonding performance of the composite material, combined with the rupture section of the composite material.

This topic meets the scope of this journal and English is good. Some good results have been obtained. It is recommended as minor revision as the following suggestions:

Suggestions:

  1. The effect of electrode arrangement direction on mechanical properties of composite with embedded sensor is not reflected in the abstract. Suggest add the result of influence of electrode arrangement direction on mechanical properties of composites.
  2. In the schematic diagram of shear specimen in Figure 4, it is suggested to mark each type in the schematic diagram of 8 types of components, so that readers can more easily understand the meaning of each schematic diagram.
  3. In the first paragraph of the conclusion, the sensor structure made of gold will reduce the interlaminar shear properties of the composite, but it does not specify the specific weakening amplitude. It is suggested to enhance the specific influence range of the sensor structure made of gold on the interlaminar bonding properties of the composite.
  4. Reference 27 has not been published and cannot be used as a reference. It is suggested to delete reference 27

Author Response

(The authors gave the same response as above.)
